# Observing eruptions of gas-rich compressible magmas from space

Brendan McCormick Kilbride[1,2], Marie Edmonds[1,2] & Juliet Biggs[1,3]

Observations of volcanoes from space are a critical component of volcano monitoring, but we lack quantitative integrated models to interpret them. The atmospheric sulfur yields of eruptions are variable and not well correlated with eruption magnitude and for many eruptions the volume of erupted material is much greater than the subsurface volume change inferred from ground displacements. Up to now, these observations have been treated independently, but they are fundamentally linked. If magmas are vapour-saturated before eruption, bubbles cause the magma to become more compressible, resulting in muted ground displacements. The bubbles contain the sulfur-bearing vapour injected into the atmosphere during eruptions. Here we present a model that allows the inferred volume change of the reservoir and the sulfur mass loading to be predicted as a function of reservoir depth and the magma's oxidation state and volatile content, which is consistent with the array of natural data.

---

[1] Centre for the Observation and Modelling of Tectonics and Volcanism (COMET). [2] Earth Sciences, University of Cambridge, Downing Street, Cambridge CB2 3EQ, UK. [3] Earth Sciences, University of Bristol, Wills Memorial Building, Queens Road, Clifton, Bristol BS8 1RJ, UK. Correspondence and requests for materials should be addressed to M.E. (email: me201@cam.ac.uk).

The vast majority of volcanic eruptions occur in remote regions and are not observed directly. Recent developments in Earth Observation capability have led to the development of a number of space-based sensors for the purposes of measuring changes in the height of the ground surface and gas emissions before, during and after volcanic eruptions. Images showing the deformation and sulfur dioxide cloud generated during the eruption of Okmok Volcano, Alaska, in 2008, for example, are shown in Fig. 1. Interferometric synthetic aperture radar (InSAR) uses differences in the phase of radar waves returning to the satellite between two or more SAR images to generate maps of surface displacement or digital elevation, typically displayed as 'fringes', which are contours of constant phase change. InSAR is particularly valuable for measuring ground deformation at volcanoes with little or no ground-based GPS networks and in addition provides spatially continuous data, allowing the size and shape of the magma source in the crust to be deduced using geometric models[1]. Observations of volcano ground deformation, however, show that some volcanoes tend to show signficant pre- and syn-eruptive deformation, whilst others do not[2]; the reasons for these variations are unclear. Sulfur dioxide clouds generated during eruptions that reach the upper troposphere or stratosphere are observed from space using spectrometers operating in the UV (for example, the Ozone Monitoring Instrument (OMI)[3]) and in the IR (IASI[4],

Atmospheric Infrared Sounder (AIRS)[5]). It is apparent that there is no clear relationship between erupted volume (dense rock equivalent, DRE) and the mass of sulfur gases emitted during an eruption[6], creating challenges for reconstructing the magnitudes and climate impacts[7,8] of past eruptions recorded as sulfate spikes in ice cores[9] and for predicting the impacts of potential future eruptions. Geochemical and geophysical observations of volcanic eruptions are becoming ever more precise, frequent and spatially resolved[10–12], yet they are rarely considered in tandem, despite the fundamental link between them.

Two important features of these data sets may be explained by the presence of a gas phase co-existing with the magma in the storage area before eruption. First, is observed that the mass of sulfur present in the co-eruptive gas cloud during explosive eruptions is not clearly related to the mass of magma erupted, nor to the eruption column height[6]. The total mass of sulfur in the cloud usually far exceeds that expected from the degassing of the erupted magmas based on the concentration of sulfur in melt inclusions[6]. This difference may be ascribed to the presence of a pre-eruptive vapour phase into which the sulfur partitions strongly[13]. Secondly, volcanic eruptions are often preceded by periods of uplift and accompanied by periods of subsidence, linked to magma intrusion, eruption and cooling and degassing of magmas[2,14]. The inferred volume change of the reservoir observed syn-eruption, however, is often many times less than the volume of magma erupted at the surface (corrected for density difference). This difference in volume is controlled by the balance between the compressibility of the magma, including that of any exsolved gas phases, and the material properties of the host rock, a feature highlighted by numerous previous studies[1,15–23]. The total mass of exsolved vapour is difficult to quantify independently however, and this lack of understanding leads to a lack of constraint on inferred volume changes from inverse modelling of InSAR observations.

The exsolved gas phase that causes the magma to be highly compressible is comprised of the primary magmatic volatile species water ($H_2O$), carbon dioxide ($CO_2$), sulfur gases ($H_2S$ and $SO_2$) and minor species involving carbon and halogens (for example, CO, HCl, HF, alkali halides). Thermodynamic saturation models for mixed volatile species are now well established, based on gas–melt and gas–gas equilibria of the type: $2CO + O_2 \Leftrightarrow 2CO_2$ and $2H_2S + 3O_2 \Leftrightarrow 2SO_2 + 2H_2O$ (refs 24–26). Sulfur partitioning in such models is constrained by experiments that quantify the abundance of sulfur in magmatic vapour in equilibrium with melts over a range of melt compositions, oxidation states and pressures[27–29], which have shown that in general sulfur partitioning into vapour increases during decompression and is heavily controlled by the relative abundance of sulfide ($S^{2-}$) over sulfate ($SO_4^{2-}$), which is in turn governed by oxygen fugacity, and melt composition[30]. Thus the abundance of sulfur in the gas phase coexisting with magma is a sensitive tracer of these parameters.

Traditionally, the magnitude of sulfur emissions into the atmosphere during explosive volcanic eruptions (measured by space-borne sensors) has been simply compared with the volume of magma erupted and its composition, to infer an 'excess sulfur' problem, whereby the mass of sulfur emitted is orders of magnitude larger than that which can be accounted for by the concentration of sulfur in pre-eruptive melts as sampled by melt inclusions[6]. Our enhanced understanding of sulfur behaviour in recent years however, has fully reconciled the observations with the presence of a pre-eruptive vapour, first proposed on the basis of observations[13] and later confirmed experimentally[31] and modelled[26]. Now a large database of sulfur emissions from volcanoes exists[5], but there have been very few attempts[20,23] to fit these observations into this now well-established theoretical

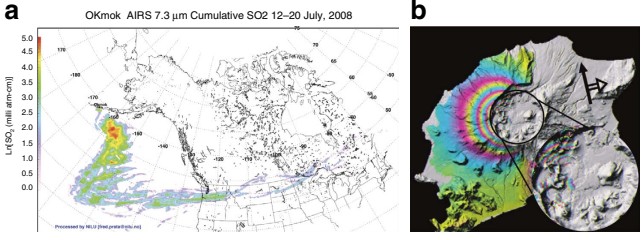

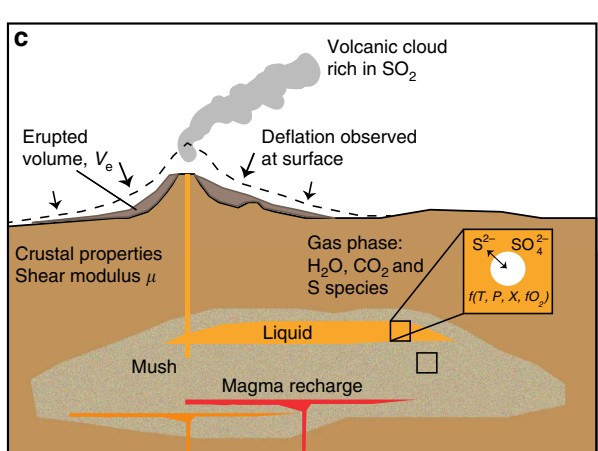

**Figure 1 | Example observations and a schematic illustration of the model underpinning this study.** (**a,b**) Observations of the 2008 eruption of Okmok Volcano, Alaska, USA. (**a**) An interferogram generated using a pair of synthetic aperture radar images for the main phase of the explosive eruption between 12 and 13 July 2008 (ref. 45) and (**b**) an AIRS image to show the spatial extent and atmospheric concentrations of the $SO_2$ cloud generated by the explosive phase of the eruption[69]. (**c**) Schematic diagram to show how sulfur partitions into the gas phase at depth, dependent on melt composition, pressure and oxygen fugacity. The gas phase causes the magma to be compressible[15], which gives rise to muted ground deformation (in direction of arrows) in response to the evacuation of the magma reservoir during eruption. The magma reservoir contains regions of liquid (orange) and crystal-rich mush, and may be recharged by mafic magmas (red).

framework for sulfur gas–melt partitioning and none reconciling the geochemical features of the magmatic vapour with the consequences for magma compressibility. This approach clearly has enormous potential for strengthening our constraints on pre-eruptive magma storage conditions.

Here we present a novel approach aimed at creating a model to reconcile satellite-based observations of surface deformation and atmospheric sulfur mass loading during discrete explosive volcanic eruptions. In doing so, the sensitivity of ground deformation and gas emissions to the intrinsic magma storage and bulk volatile contents of the magma will be explored.

## Results

The model combines a thermodynamic model of volatile saturation and partitioning[25], which describes the composition of a gas phase in equilibrium with melt as a function of volatile content, melt composition, temperature, pressure and oxidation state, with a physical model describing the magma compressibility effects caused by the exsolved gas[19], illustrated in Fig. 1c. The model predicts sulfur mass loading and the magnitude of the reservoir volume change as inferred from geodetic observations at the surface. The magnitude of the reservoir volume change, which may be compared with the erupted volume (DRE) using the notation $r$, which may be defined as the ratio between the erupted volume ($V_e$) and volume change of the subsurface reservoir ($\Delta V_c$), given by[15] $r = V_e/\Delta V_c$. Three theoretical possibilities can be envisioned for a spherical source: firstly, if the magma is considered incompressible, the reservoir volume change would be equal to the erupted volume ($r = 1$). Secondly, for a gas-free but slightly compressible magma, the reservoir volume change will be less than the erupted volume ($r > 1$). Thirdly, an exsolved volatile phase in the magma will increase its compressibility by an order of magnitude, thus the eruption of a significant volume of material is accommodated by expansion of the remaining magma causing relatively little volume change of the reservoir[16–18,32] ($r \gg 1$). We note that assumptions about geometry and material property are required to infer subsurface volume change from the surface displacements observed by geodetic techniques such as GPS and InSAR. As our goal is to establish a framework for reconciling observations of gas and deformation, we consider the simplest geometry—that of a spherical source—and use derived values of volume change from published studies. It is our intention that future studies refine this framework for specific circumstances, and the influence of these assumptions are discussed in Supplementary Material.

The model predicts the sulfur mass loading, $M_S$ and change in reservoir volume, expressed as a ratio $r$, to the erupted volume based on a set of input parameters (total $H_2O$, total $CO_2$, oxidation state, melt composition), which may then be compared with space-borne observations (Fig. 1; Supplementary Table 1). The results of the model for a generalized metaluminous rhyolitic magma composition are shown in Fig. 2. Partitioning of sulfur into vapour in the pre-eruptive magma reservoir varies over orders of magnitude for the conditions studied (particularly upon varying the water content of the melts and the magma's oxygen fugacity) and in all cases decreases with depth, with partition coefficients comparing well to those from experimental studies[30,31]. As a consequence, the mass of sulfur expected to be present in the co-eruptive atmospheric cloud per $1 \text{ km}^3$ magma ranges over several orders of magnitude, as is also observed for natural eruptions (Fig. 3; Supplementary Table 1). The $H_2O$ and $CO_2$ contents of the magma, as well as the oxidation state, are all important for controlling the partitioning of sulfur into a vapour, or gas phase in the crustal magma reservoir before eruption and hence determining how large (in mass) the co-eruptive sulfur

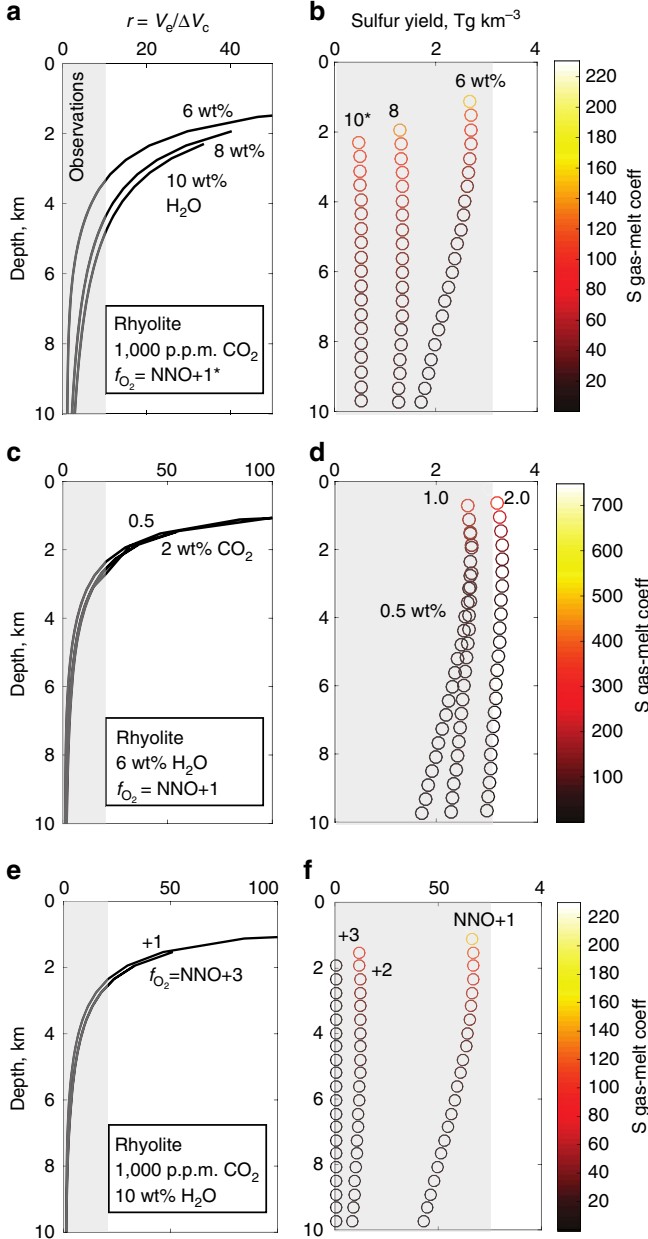

**Figure 2 | Model to show the effects of varying magma storage conditions on the magnitude of the observed ground deformation and the atmospheric sulfur yield of eruptions.** Plots to show the effect of (**a**,**b**) bulk magma water content, (**c**,**d**) bulk magma $CO_2$ content and (**d**,**e**) magma oxidation state on $r$ (erupted volume divided by inferred change in reservoir volume; $V_e/\Delta V_c$ (**a**,**c**,**e**)) and the sulfur yield (Tg km$^{-3}$ magma erupted (**b**,**d**,**f**)). NNO is an abbreviation for Ni–Ni–O buffer, a measure of relative oxygen fugacity. *In **a**, the oxygen fugacity is NNO + 1 for the model runs with a melt water content of 6 and 8 wt% $H_2O$ and NNO + 2 for the model run with a melt water content of 10 wt%, justified by global observations[70]. The sulfur yields with depth are colour-coded for their sulfur vapour-melt partition coefficients. The observed range in sulfur loadings and $r$ values for observed volcanic eruptions (Supplementary Table 1) are shown as gray shaded boxes.

cloud that is observed from space is. In particular, the oxidation state of magma exerts a primary control on sulfur exsolution, owing to the much higher solubility of the sulfur existing as $SO_4^{2-}$ over $S^{2-}$ (ref. 33). In contrast, the melt $H_2O$ content is the main control on magma compressibility; the effect of oxidation

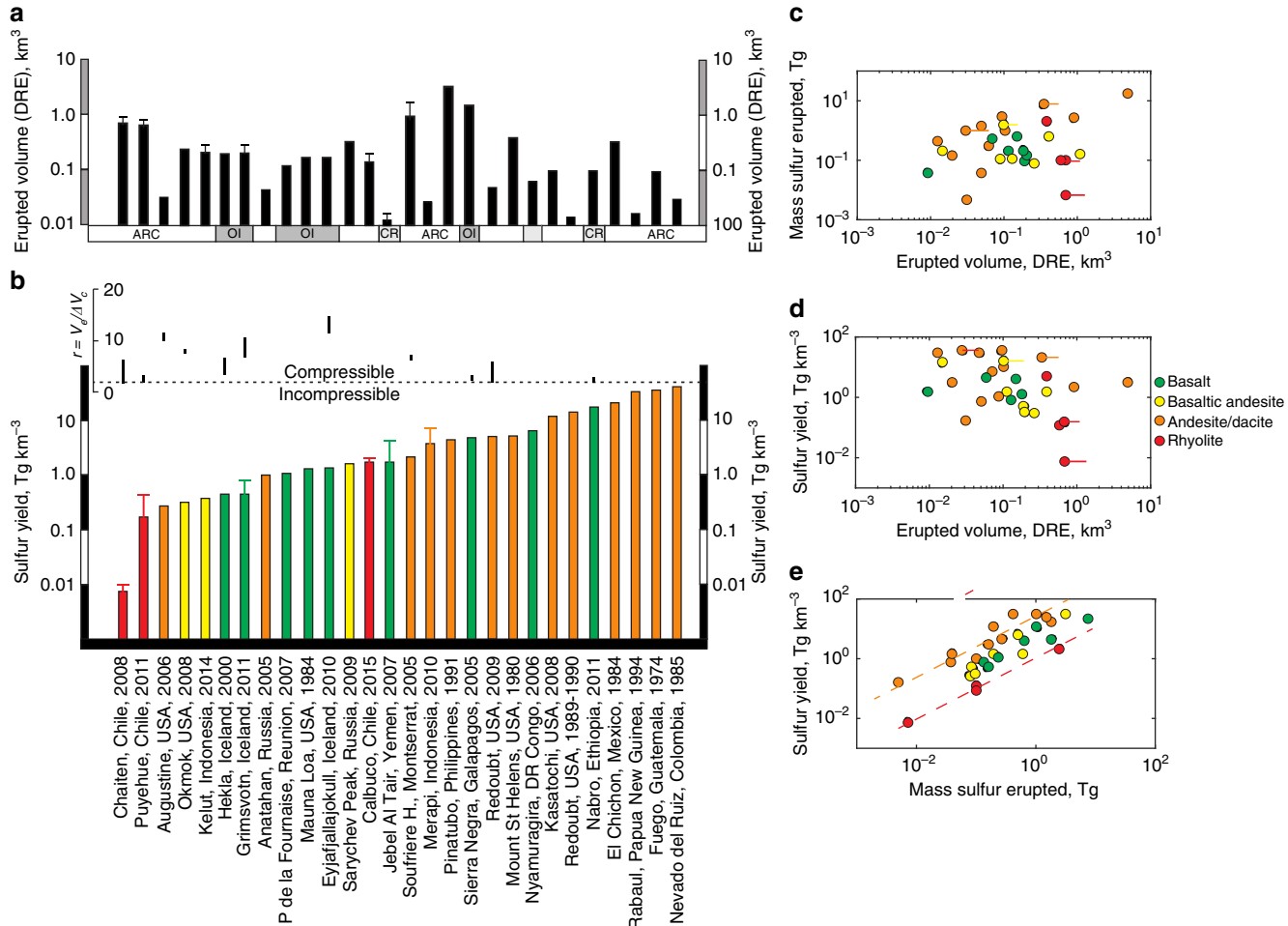

**Figure 3 | Sulfur output and erupted volume systematics for natural volcanic eruption data.** (**a**) Erupted volume for selected eruptions (dense rock equivalent, DRE; km$^3$). CR, continental rift; ARC, subduction zone-related volcano; OI, ocean island volcano. (**b**) Column plots of sulfur yield (Tg km$^{-3}$) for the eruptions in **a** (listed in Supplementary Table 1). Columns are colour-coded for magma composition (legend far right). Vertical bars above the columns show $r$ values (erupted volume divided by inferred change in reservoir volume; $V_e/\Delta V_c$) for the eruptions for which these data are available. The deformation data are derived from InSAR studies as well as ground-based GPS networks. Uncertainties in the observations, where available, are shown as vertical error bars. (**c–e**) Correlations between eruption size and sulfur yield, all data points colour-coded for magma composition. (**c**) Mass of sulfur erupted versus erupted voume (DRE; km$^3$). (**d**) Sulfur yield (Tg km$^{-3}$) versus erupted volume (DRE; km$^3$). (**e**) Sulfur yield (Tg km$^{-3}$) versus the total mass of sulfur emitted (Tg).

state is negligible (this is because sulfur makes up such a small proportion of the gas phase). The model also allows prediction of melt volatile contents and the composition of volcanic gases in equilibrium with the magma pre-eruption (Supplementary Fig. 1).

The ratio $r$, which is proportional to magma compressibility (via equation (2), Methods), is predicted to increase with decreasing magma reservoir depth from a minimum of around 1.05 (governed by crustal properties, see methods) up to a value of >40 for magmas stored at depths of <2 km before eruption (Fig. 2). For magma reservoirs at 3–4 km in the crust, the $r$ value is predicted to be 10–20 (assuming no gas loss). Magma compressibility scales with the first derivative of density with respect to pressure (equation (1), Methods), and so $r$ is largest for eruptions tapping magma from the shallowest parts of the system. While the value of $r$ is not affected significantly by the bulk $CO_2$ in the system, or by magma oxidation state (Fig. 2), it is affected significantly by the bulk magmatic $H_2O$ content. Increasing the magma $H_2O$ content from 6 to 10 wt% increases $r$ from 2 to 7 for a magma reservoir at 6 km, and from 7 to around 14 for a magma reservoir at 4 km depth (Fig. 2). For magma reservoirs deeper

than around 8 km, the $r$ value is predicted to be close to the incompressible case ($\sim1.4$) for all but the most water-rich magmatic systems.

**Comparison to natural data for volcanic eruptions.** Data for sulfur yields (outgassed sulfur divided by erupted volume (DRE) and $r$ values (erupted volume DRE/volume change of modelled source from deformation data) are shown in Fig. 3 and Supplementary Table 1. Where the eruptions transition from an initial explosive phase into a longer-lived effusive phase, as is common, we use data for the initial explosive phase only, such that the observed deformation and gas cloud is the result of a short-lived discrete event, therefore maximizing the possibility that the two signals are causally linked. Observations of volcanic eruptions yield $r$ values of up to a maximum of around 14 and sulfur yields (sulfur outgassed per km$^3$ DRE magma erupted) up to a maximum of 34 Tg km$^{-3}$ but more commonly up to around 5 Tg km$^{-3}$ (Fig. 3; Supplementary Table 1) (equivalent to 2,000 p.p.m. sulfur in the bulk magma). These apparent maxima in 'natural' sulfur yields and in the ratio $r$ may be related to the

coupling of the magma and the exsolved vapour phase in the magma storage region before eruption. The porosity of magma with 6 wt% $H_2O$ and 0.1 to 1 wt% $CO_2$ is predicted to reach 25–40% for an $r$ value of 10 (Supplementary Fig. 2). For isobaric, crystal-free magma the 'percolation threshold' occurs at a porosity of 30%, at which point uniform spherical bubbles begin to overlap[34]. The resulting increase in permeability allows the gas to escape, perhaps placing a limit on the proportion of gas in the reservoir and hence on values of $M_S$ and $r$. The presence of significant amounts of crystals in the magma is expected to modify porosity–permeability relationships further. Experiments using analog materials have shown that crystals significantly alter the bulk rheological properties of magmas, leading to a range of gas transport behaviours involving fingering and quasi-brittle fracturing[35,36].

There are only a handful of eruptions for which both the sulfur yield and the $r$ value has been measured or observed (Figs 3 and 4; Supplementary Table 1). Within these natural data there are some significant trends. The sulfur yield (in $Tg\,km^{-3}$ magma erupted) shows a positive correlation with the total sulfur emitted (Fig. 3b). Within these data, the sulfur yields for andesites appear to elevated over those for rhyolites, consistent with previous studies[6]. In contrast to previous studies[6], our data show no clear relationship between erupted magma volume (DRE) and the mass of sulfur emitted syn-eruption. From our data set of recent eruptions, the Nabro eruption gave rise to one of the largest sulfur yields (18 $Tg\,km^{-3}$ magma); and Chaiten the lowest (0.008–0.01 $Tg\,km^{-3}$). These features of the data may be explained by some eruptions tapping a zoned magma reservoir, where exsolved gases have accumulated at the top of the magma body, giving rise to extremely gas-rich eruptions, as suggested by Wallace and Carmichael[37]. Alternatively, there may be other first order thermodynamic controls on the pre-eruptive exsolution of sulfur into vapour from magmas, as we show in Fig. 2. It is likely that both processes are important and we show the vectors corresponding to the effects of these factors on both the sulfur yield of eruptions and on the ratio $r$ in Fig. 4.

The sulfur yield of an eruption is controlled by the accumulation of gases at the topmost portions of the magma reservoir, as recently suggested for Soufrière Hills[38], the bulk volatile content of the magma, and the oxidation state. Highly oxidized magmas, which may saturate anhydrite early[33,39], are likely to produce sulfur-poor vapour (or gas) and sulfur-poor eruption clouds (Fig. 4)[31,40]. Increasing the magma $CO_2$ content from 0.1 to around 1 wt% has the effect of more sulfur partitioning into the gas phase. Inherently sulfur-rich melts (high $f_{S_2}$) or reduced magmas (low $f_{O_2}$) will tend to saturate sulfide early and hence remove sulfur available for outgassing[41]. The change of reservoir volume, and hence surface deformation, is expected to be most muted (that is, high $r$) for shallow magma reservoirs, for very water-rich magmas, or for gas-rich magma reservoir caps[15,21]. The deformation response will be strongest (low $r$) for magma reservoirs that are poor in exsolved gases and magmas that are poor in $H_2O$ (Fig. 4). The plot of $r$ value against sulfur yield for explosive volcanic eruptions (Fig. 4) shows that in general, as expected, the shallower magma reservoirs give rise to lower sulfur yields and higher $r$ values (that is, more muted deformation). Conversely, deeper magma reservoirs generally give rise to higher sulfur yields and changes in source volumes which match the erupted volumes. The effects of a more or less compliant chamber (determined by the crustal shear modulus and reservoir geometry) would tend to decrease or increase $r$ respectively, whilst not affecting the sulfur yield of the eruption, but insufficient data are currently available to test this aspect of our theory. There is a hint that crystal-free rhyolitic eruptions in arcs produce low sulfur yields and low $r$, andesitic eruptions in

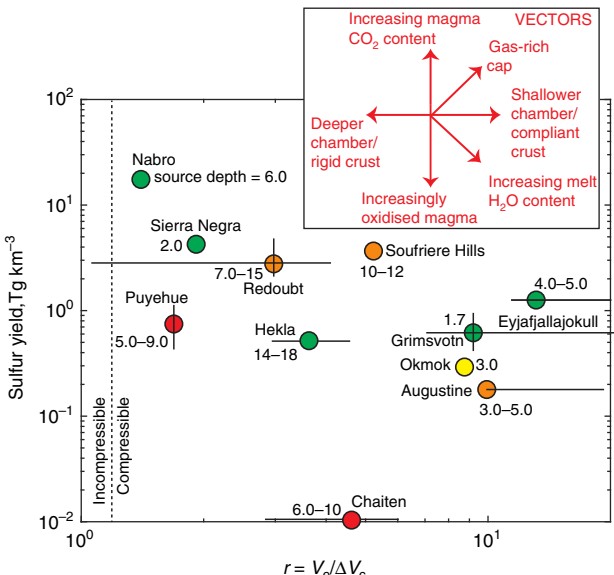

**Figure 4 | Factors controlling atmospheric sulfur loading and ground deformation systematics for volcanic eruptions.** The sulfur yield ($Tg\,km^{-3}$) is plotted against the ratio $r$ (erupted volume/observed volume change of a modelled source, from ground deformation). Colour code indicates erupted magma composition, where red: rhyolite; orange: andesite; yellow: basaltic andesite and green: basalt. The depth of the modelled ground deformation source for each eruption is shown next to the data point (in km). Vectors to show the effects of varying magmatic and crustal parameters are shown.

arcs produce high sulfur yields over a range of $r$ and basaltic eruptions in a range of settings display diverse behaviour.

**Case studies.** We illustrate the combined modelling approach using two case studies from Fig. 1; the first representing perhaps the standard case, the second an end member, illustrating the range of possible behaviour. An explosive eruption occurred at Okmok Volcano on 12th July 2008 which ejected 0.15 Tg $SO_2$ (refs 42,43) into the stratosphere, which was tracked across North America[44]. The eruption was followed by a period of subsidence[45]. A total of 0.26 $km^3$ (DRE) magma was erupted over the course of the eruption in the form of ash-rich explosions[43]. The erupted magma was a crystal-poor basaltic andesite bearing plagioclase, olivine and clinopyroxene phenocrysts, thought to have been remobilized by an intruding, more mafic magma[43]. The magma reservoir was inferred to be at around 3 km depth in the early stages of the eruption, with deeper levels, up to 6 km, evacuated later in the eruption (from inversion of InSAR data[45]), which is supported by melt inclusion $H_2O$ concentrations, which range from 0.6 to 3.6 wt% (ref. 43). The $r$ value (8.7; Fig. 1, Supplementary Table 1) and the sulfur yield (0.29 $Tg\,km^{-3}$) for the initial 13 h of the eruption are consistent with magma being evacuated from a storage region at 3 km (Supplementary Fig. 3A), with a magma oxidation state of $NNO+1$, a bulk $H_2O$ content of $\sim$3–4 wt% and bulk $CO_2$ content of 0.5–1.0 wt%, which is a reasonable range for arc magmas[46]. These pre-eruptive magma conditions can explain both the magnitude of the deformation signal as well as the sulfur yield. This analysis takes no account of sulfur supplied from intruding mafic magma.

The Chaiten eruption, in Chile in May 2008, was a rhyolitic plinian eruption with a VEI (volcanic explosivity index) of 4 and an eruption column that reached 19 km above sea level during a sequence of explosive events over six days[47]. Geodetic (InSAR)

data constrained the magma reservoir to be a sill-shaped body at a depth of $>6$ km (ref. 48). The sulfur burden to the atmosphere associated with these eruptions was small, only 0.01 Tg $SO_2$ (ref. 49). Erupted rhyolitic pumices were very crystal poor[50]; the very low crystallinity prevents constraints on the magma volatile content from melt inclusion data, but petrological experiments suggest that the magma was saturated with respect to $H_2O$ at pressures of up to $\sim 190$ MPa (ref. 50). One explanation for the very sulfur-poor gas cloud accompanying the eruption might be a particularly oxidized magma (Fig. 2). We find that the constraints provided by the difference between the modelled volume and erupted volumes (an $r$ value of 2.8–6.0), the low sulfur burden[49], a magma reservoir depth of $>6$ km (ref. 48) and a vapour-saturated magma[50] may be satisfied by magma containing a bulk $H_2O$ content of $>7$ wt% (with a $CO_2$ content of 0.1 wt%) and an oxidation state 2–3 log units above the NNO buffer (Supplementary Fig. 3B). A high $f_{O_2}$ may cause the vapour-melt partitioning behaviour of sulfur (as $S^{6+}$) to be severely damped[30,31,40] resulting in a low sulfur mass in the ejected gas cloud (which would have been mostly comprised of water vapour). The high magmatic water content would have yielded a large $r$ value despite the relatively deep magma chamber (Supplementary Fig. 3B). The model used to calculate the subsurface volume change for eruption is that of a sill[48], a geometry that is significantly more compressible than a spherical source used by our proposed model (see Methods and Supplementary Material for discussion)[51]. The inferred reservoir volume change and value of $r$ are only slightly affected by this assumption, but there is an inherent tradeoff between magma and reservoir compressibility in equation (2). Thus, to achieve an equivalent value of $r$, a more compressible chamber requires a more compressible magma, and it is likely that we actually underestimate the magma compressibility and hence the magmatic water content. Alternatively, the low sulfur yield may be due to either (a) an inherently sulfur-poor primary melt, which seems unlikely on the basis of global trends[52] or (b) a lack of a pre-eruptive magmatic vapour phase (much of the sulfur would be expected to reside in the vapour phase at lower oxygen fugacities, $\sim$ NNO $+ 1$ (ref. 27)), but this is unlikely on the basis of the muted deformation signal observed (high $r$; Figs 3 and 4).

## Discussion

Our approach is the first to attempt to reconcile diverse global geophysical and geochemical observations of volcanic eruptions from space and is intended to be an initial framework for enhanced understanding, providing a basis from which to improve and diversify volcanic eruption modelling. There are of course caveats to this study: the magnitude and time-dependency of volcano deformation may be influenced by the thermal state of the crust and any viscoelastic component of deformation; deformation arising from dyke emplacement and ascent will be superimposed onto the deflation signal accompanying magma withdrawal from a deep reservoir; magmas might outgas sulfur (and other species) during ascent to the surface, which will contribute sulfur to the eruption cloud that is not linked to the volume systematics. Magmas might outgas sulfur but not erupt; this would lead to higher sulfur yields per unit volume of magma erupted. We have attempted to minimize these aspects by focussing on discrete explosive eruptions which have short and well defined deformation signals and which involve rapid magma decompression and explosive eruption into the upper troposphere or stratosphere. As observations become better constrained and measurements more precise we anticipate that integrated modelling of this kind will become commonplace and there is scope for additional layers of complexity; in particular, the

presence of macrocrysts and plumbing system architecture will have a profound influence on gas transport and storage in magmatic mushes and reservoirs which is not yet well understood.

## Methods

**Thermodynamic modelling.** The model DCompress[25] is used to generate gas compositions in the system C–S–O–H–Fe during decompression of a melt where, for any volatile species dissolved in a vapour-saturated silicate melt, equilibrium conditions impose that the fugacity of species in the gas phase equals that in the melt[53], established using mass balances and the equilibrium constants of the reactions occurring in the gas phase[26]. The dissolved amounts of the soluble species are determined using solubility laws that are a function of the corresponding species fugacities[25]. The equilibrium constants for a set of redox reactions involving H–O–S–C are calculated using established thermodynamic data[54]. For decompression of magma, at each pressure increment the proportion of gas, the composition of gas and melt with respect to the volatile species $H_2O$, $CO_2$, $CO_2$, $H_2S$, $SO_2$ are calculated and the melt, gas and bulk densities. The total moles of gas in the vapour and the volume fraction of the gas phase and the mass fractions of sulfur in the gas and melt phases during closed system degassing may be calculated and from these, the vapour-melt partition coefficient for sulfur ($S_{fluid-melt}$). Combining the mass of sulfur in the gas with estimates of erupted magma mass (from field-based estimates) leads to estimates of total sulfur released during explosive eruption ($M_S$), which is equal to the mass of sulfur present in the pre-eruptive gas phase in addition to the mass of sulfur liberated during decompressional degassing.

Also arising from the gas phase calculations are a mass of vapour per unit volume of magma. Using the molar proportions of $H_2O$, $CO_2$ and S gases in the vapour combined with their molar masses, it is possible to calculate the total mass of vapour associated with 1 m³ magma at pressure increments. The molar volumes of the vapour may then be used to calculate a bulk magma density, $\rho$, at each pressure step, using the Ideal Gas Law and the bulk density may be calculated using an appropriate density for the melt proportion (2,400 kg m⁻³ for rhyolite; 2,800 kg m⁻³ for basalt).

**Compressibility of the gas phase.** The magma compressibility, $\beta_{magma}$, including that arising from this sulfur-bearing gas phase, is given by[19]:

$$\beta_{magma} = \frac{1}{\rho}\frac{\partial \rho}{\partial p} \qquad (1)$$

where $\beta_{magma}$ is the bulk magma compressibility (Pa), $\rho$ is magma density (from the outputs of the thermodynamic model, described above) and $p$ is pressure. The volume removed during eruption is accommodated by deformation of the host rock and compressibility of the magma. Therefore, the ratio, $r$, between the erupted volume ($V_e$) and the change in volume of a magma reservoir (the 'Mogi' source[55]) ($\Delta V_c$) is given by[15]:

$$r = \frac{V_e}{\Delta V_c} = 1 + \frac{\beta_{magma}}{\beta_c} \qquad (2)$$

where $\beta_c$ is the chamber compressibility and $\beta_c = \frac{3}{4\mu}$ (where $\mu$ is the shear modulus of the host rock[17]) for a spherical chamber. The compressibility of a spherical cavity is always less than that for an ellipsoidal cavity. For deep prolate sources, $\beta_c = 1/\mu$, so the difference in compressibility for these cases does not exceed 25%, but for extremely oblate cavities, approximating a circular crack, the difference can be two orders of magnitude[21,51]. This contrast in compressibility has been applied to dyke intrusions to explain the discrepancy between the volume change of the source and sink[15,51].

The shear modulus of crust, $\mu$, in volcanic areas is not well known and ranges from $\sim 0.1$ GPa for very compliant rocks to 30 GPa for non-compliant, or stiff rocks[15,56]. Compressibility of degassed basalts, $\beta_{magma}$, at crustal depths is in the range $0.6–1.0 \times 10^{-10}$ Pa⁻¹, resulting, in theory, in a value of $r$ for degassed magma ranging from 1.05 to 9, which encompasses much of the range in the natural data. Improved constraints on the compliance of the crust are clearly needed to refine these models in the future. An equally or perhaps more important control on the range in $r$ is the variability in the value of $\beta_{magma}$, which we predict will extend up to $30 \times 10^{-10}$ Pa⁻¹ for volatile-rich magmas stored at depths of $>3$ km, yielding a large range in $r$, up to 15 for realistic crustal parameters and magma depths.

**Uncertainties in satellite-based observations of $SO_2$ clouds.** The observations of gas clouds collated for this study are from a number of different sensors on different platforms, but are dominated by observations by the OMI and the AIRS[5], which quantify vertical column densities of sulfur dioxide using its characteristic absorption signature in the UV and IR, respectively[57,58]. The observation of $SO_2$ from space using radiance data is subject to many sources of error and uncertainty[5,59], which are summarized here. They may be grouped into uncertainties arising from sensor response (including the OMI 'row anomaly'[60]), retrievals (for example, interference with other gases, principally ozone[58]), local factors (for example, cloud cover, interference by ash[61]), detection limits[3], inherent

features of plumes such as $SO_2$ loss by oxidation and dilution[59]. In this study we have attempted to minimize errors by focussing only on larger, short-lived, discrete explosive eruptions, where gas plumes are injected high into the troposphere and even into the stratosphere. Absolute errors are difficult to constrain, as other ground-based techniques to measure $SO_2$ are likewise subject to errors including those caused by excess scattering of UV radiation[62] but attempts have been made to conduct calibration and comparison experiments using opportunistic observations of gas clouds from the ground and from space[44,63].

**Uncertainties in thermodynamic modelling.** The equilibrium constants for the redox reactions in the gas phase in the model Dcompress[25] are calculated after[54]. The solubility laws are constrained experimentally and are given in Table 1 of Burgisser et al.[25] The uncertainties within the model arise from exploring the less well constrained parts of the melt–C–O–H–S fluid solubility regime. In the examples we investigate here, we explore standard basaltic and rhyolitic melt compositions, for which there are numerous experimental studies of volatile solubility to draw on, for example, Lesne et al.[28] and thus the model is well constrained for these compositions. Estimates of the uncertainties on melt volatile compositions are in general $<10\%$ for pressures $>1$ kbar (ref. 25).

**Uncertainties in geodetic estimates of volume change.** The observations of volume change in this study are collated from both satellite- and ground-based measurements of surface deformation, both of which are subject to inherent limitations and uncertainties. The ability of InSAR to measure surface deformation is limited by the availability of satellite images and the correlation between the images (the coherence of the interferogram). Coherence is limited by steep slopes, particularly for image pairs with high perpendicular baselines, and surface changes, which are particularly common in vegetated regions, for image pairs with long temporal separation and for interferograms created using short-wavelength instruments (X-band, C-band)[64]. The uncertainty of the deformation measurement is dominated by the influence of tropospheric water vapour, and is particularly an issue for high relief volcanoes where atmospheric stratification causes systematic error correlated to topography[65]. Under favourable conditions, long, dense timeseries of InSAR data can give uncertainties better than 1 mm per year[66], but for a single interferogram at tropical volcanoes, errors can be as much as 17 cm (ref. 67). For ground-based networks such as GPS, the limitation is typically the number and location of instruments, particularly during large explosive eruptions as near-field sensors are often inoperable or destroyed.

The greatest source of uncertainty is the assumptions required to convert surface deformation into source volume changes. For simplicity, this paper is based on spherical sources within uniform, elastic half-spaces[55], but in practice, many more parameters need to be considered, in particular topography and thermal and mechanical heterogeneities[68]. Of particular relevance here, is the shape of the reservoir, which controls chamber compressibility. The compressibility of a spherical cavity is always less than that for an ellipsoidal cavity. For deep prolate sources, $\beta = 1/\mu$, so errors do not exceed 25%, but for extremely oblate cavities, approximating a circular crack, the difference can be two orders of magnitude[21,51]. This contrast in compressibility has been applied to dyke intrusions to explain the discrepancy between the volume change of the source and sink[15].

**Data availability.** All of the data used in this paper have been published in the peer reviewed literature and are given in the Supplementary Material, table 1, which also contains the primary references for each data set.

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

## Acknowledgements

The authors gratefully acknowledge the support of the Deep Carbon Observatory and DECADE (part of the Reservoirs and Fluxes community), the Natural Environment Research Council (NERC) Centre for the Observation and Modelling of Earthquakes, Volcanoes and Tectonics (COMET), the British Geological Survey and the University of Cambridge Isaac Newton Trust. We thank Paul Segall and two anonymous reviewers for providing reviews which improved the paper enormously.

## Author contributions

M.E. and B.M.K. conceived of the idea. J.B. formulated the equations for the ground deformation modelling. B.M.K., J.B. and M.E. collated the data. M.E. performed the calculations, plotted the data and wrote the manuscript. All authors discussed the results and edited the manuscript.

## Additional information

**Competing financial interests:** The authors declare no competing financial interests.

