## [Peer Review File · Nature Communications]

Reviewer #1 (Remarks to the Author):

The paper describes a new approach to understand the occurrence and variation of gas-rich eruptions by linking two kinds (passive and active) of satellite observations with magmatic processes. The paper is clearly written and contains new material, presented in an informative and rigorous manner. The observation of very rich or poor yields of SO₂ gas from explosive volcanic eruptions has not previously been explained or investigated in an entirely satisfactory way. This paper provides some new insights into gas compressibility in the magma and a model that helps to understand the wide variation of SO₂ yields. The authors are careful to caveat their results and do not claim more than they can from the analysis of the data presented. This appears to be the first paper suggesting an approach that utilises satellite data in a quantitative manner to elucidate processes deep below the surface.

The model approach is interesting because it uses the observations to try to delineate magmatic processes without taking a formal mathematical approach of assimilating data or using mathematical inversion. I would suggest the more formal approach is premature as the data are not sufficiently accurate and the models are under development.

This paper is a very timely and a novel contribution - it has all the hallmarks of originality and will be of great interests to those studying explosive volcanic activity. The authors themselves note that this work is a "first attempt" and I would anticipate that refinements will follow rapidly.

The data and methodology are sound. The quality of presentation is also excellent and the Supplementary material I found very helpful.

There is not much discussion on uncertainties and I find this the only weakness in the paper. I think some statements on the uncertainties in model parameters and especially concerning the satellite data observational errors would be worthy of inclusion - perhaps in the Supplementary material(?)

The conclusions are clear and robust and not extravagant. I have a feeling that the authors have a lot more interesting material to present and this is at an early stage of development. I assume that other papers are in preparation but that the need to publish their ideas quickly is paramount. The paper is therefore, in my view, very appropriate for Nature Communications.

References are appropriate and up-to-date (at least in the area of research that overlaps with mine). I would suggest one additional reference:

Kremser, S., et al. (2016), Stratospheric aerosol—Observations, processes, and impact on climate, *Rev. Geophys.*, 54, doi:10.1002/2015RG000511.

This is a new (review) paper and links nicely with this work because it shows why we need to know more about how and why volcanoes sometimes emit large amounts of SO₂ and why they are sometimes gas-poor.

Reviewer #3 (Remarks to the Author):

Review of "Observing eruptions of gas-rich, compressible magmas from space", by McCormick-Kilbride et al.

In this work, the authors compute the compressibility ratios 'r' for a number of eruptions using estimates of erupted volume and reservoir volume change from the literature. Using the solubility package D-Compress, and making assumptions about reservoir compressibility, magma compressibility is related to volatile concentrations. Assuming degassing during eruption, these estimates are related to observations of sulfur emissions.

This work is a valuable contribution to the literature and the authors are to be commended for working to integrate disparate observations of volcanic eruptions. Quantitatively linking observed gas emissions with subsurface processes is an important problem and should inspire future work. There are however some important issues that need to be addressed prior to publication. These issues are described below.

General Comments

=====

(1) I do not believe that Equation (1) is correct. The compressibility of a multiphase melt may be computed as the sum of the individual compressibilities multiplied by their respective volume fractions (see for instance Mastin et al. 2008, equation 6). Equation (1) here shows the bulk magma compressibility as the sum of melt (gas-free) and gas phases, but does not account for their relative abundances. The cited reference for this expression is Huppert and Woods 2002, but that paper (Equation 5) actually shows a sum of rock and magma compressibility, which is not the same as summing individual phase compressibilities in the magma. Since gas volume fractions are much lower than melt volume fractions, it seems this error could have a significant influence on results presented in the manuscript.

(2) There are some problems with the way that reservoir compressibility β_c is handled in this work. First, there is a minor confusion on Line 283, which describes β_c as host rock compressibility, when it is in fact reservoir compressibility (these are not the same). More importantly, chamber compressibility is given as $3/4\mu$. However, this expression is only strictly correct for a spherical reservoir in a full-space. In other cases, compressibility deviates from $3/4\mu$. Analytical expressions for certain end member cases can be found in Amoruso and Crescentini (2009), and numerical computations for chambers of different shapes and depths can be found in Anderson and Segall (2011). For shallow, sill-like chambers, results can be very different than $3/4\mu$. Interestingly, an expression for an opening dyke is given on Line 284, but it is not at all clear how this expression is used in the work. Geodetic references used in this study should be consulted to ensure that the assumption of a near-spherical, relatively deep reservoir is actually valid -- or, if the expression for an opening dike is used, that must be clarified. Finally, some accounting must be made of the great deal of uncertainty associated with host rock rigidity, to which reservoir compressibility is directly related. Altogether, the authors seem to assume that reservoir compressibility is known, when in fact it may be only very poorly known. At the least, some discussion about this assumption is warranted. Clearly, for a very bubbly magma the reservoir compressibility will not be important, but the minima of 1.4 mentioned in the text could certainly be affected.

(3) There are some problems with the way deformation modeling is handled, or at least discussed. First, the relationship between observed deformation, reservoir volume change, and estimates of 'r' are a bit unclear. As far as I can tell, surface volume change and reservoir volume change are being confused, at the very least in the text (it's not clear if computations are influenced). For instance, Line 58 refers to "the magnitude of the subsidence signal", while just a little later on Line 69 it is "volume change of the chamber at depth" that is discussed. The ratio r is computed from equation 2 using reservoir volume change, not surface volume change. Although volume of surface uplift or subsidence (integrated vertical displacements) is also a metric used in deformation studies (such as the Johnson 2000 paper referenced here), it is not generally the same as reservoir volume change. Also, later, on Line 278, ΔV_c is defined as "change in volume observed by InSAR". It is important to recognize that ΔV_c is change in *reservoir* volume, and it is not observed by InSAR, but is rather modeled from InSAR data. These points need to be clarified in the text, and the meaning of ΔV_c more clearly stated.

I also found the discussion on lines 62-70 a bit problematic. These statements about compressibility are well known, and assertions in i--iii need to be backed up with either specific citations or equations. Also, I suspect that reference 14 (Mastin) is not the one intended.

A final point is that the manuscript focuses on the use of space-based observations, but some geodetic inversions utilized by the cited references likely include other observations as well (for instance, the Sigmundsson paper also uses GPS data to derive subsurface volume change).

(4) The novelty of magma/reservoir compressibility and constraining a reservoir's exsolved volatile content is, I feel, a bit over-stated. The abstract states that "most eruptions are associated with ground deformation much less than expected for the erupted volume", but this is not really true -- the difference between erupted volumes and inferred reservoir volume change, and relation to magma compressibility, has been well known for quite some time (in other words, no one should really expect these things to generally be equal). Some discussion of previous work here is warranted (references include Johnson 1987, 1992, and 2000, Rivalta and Segall 2008, etc.); also, Line 60 should include some references to previous examples of the phenomenon being discussed. Previous workers have also directly modeled reservoir volatile content using the ratio r , as done here. For instance, both Mastin et al 2008 and Anderson and Segall 2013 use erupted volume and ground deformation data at Mount St. Helens, together with H₂O + CO₂ solubility models, to estimate exsolved water and CO₂ in the reservoir. This is precisely the approach followed in this manuscript, with the additional incorporation of SO₂ emissions data and a model that includes sulfur solubility.

(5) As noted in the manuscript, reservoir compressibility is dominated by exsolved water, not SO₂. Thus, estimates of r yield mostly constraint on water content. Constraint on sulfur, CO₂, and NNO must come from solubility modeling and observed SO₂ emissions. This would seem to be a difficult problem, particularly when considering the common occurrence of "excess" sulfur in the reservoir. In other words, given all these adjustable parameters, isn't it perhaps almost too easy to reproduce observations? I understand that in the case of very low sulfur emissions (Chaiten), the lack of excess sulfur simplifies the problem, but what's less clear to me is what can be learned in the more common case (Okmok). For instance, it's noted that the given parameters are reasonable for an arc magma -- but would "unreasonable" parameters also fit the data? Some further discussion would be very helpful here.

Also, Figure 2B shows erupted sulfur vs. sulfur yield, and the text notes some correlation. But isn't sulfur yield computed directly from erupted sulfur and the erupted volume? In that case, is the correlation observed not simply related to the erupted volume ($T_g/km^3 / T_g = 1/km^3$)? Am I missing something?

(6) Some clarification of some of the modeling work would be very helpful. First, are sulfur emissions set as equal to the exsolved phase only? Lines 265-269 seem to imply that these emissions are included, but Lines 241-243 seems to imply that they are not. Also, for Figure 3A, can you explain why sulfur yield decreases with increasing water content? Since D-Compress is a bit of a black box, this relationship is not immediately obvious (to me, anyway). I also find Figure S3 quite confusing. Is the idea that only certain combinations of NNO, H₂O, and CO₂ fit the data? That's great, but I'm confused by the dashed lines vs. solid dots.

Other comments

=====

Line 21: A bit unclear. How can you predict surface deformation and other observations without knowing the volume of erupted products, etc.? Certainly, reservoir depth, oxidation state, and volatile content are inadequate to predict surface deformation.

Line 33: "typically displayed as 'fringes'" is potentially confusing in the main text. Consider removing or moving to the appropriate caption.

Line 37: I think there is a big difference between a pre- and syn-eruptive lack of deformation. In

the latter case, material is clearly leaving the reservoir. But it is not necessarily clear that magma is entering the reservoir prior to eruption.

Lines 62-70: I find these three possibilities a bit confusingly described. Can this be simplified?

Line 71: Modeling of deformation data yields volume changes, but pressure change cannot be inferred independently of volume. Is the inference here that pressure change can be inferred from the solubility modeling? If so, please clarify and/or expand this.

Line 95: "there have been very few attempts". Please provide citations.

Line 97: "even fewer reconciling". Please provide citations.

Lines 101-106: I don't really understand the difference between these two items. How can you explore the sensitivity if you don't have a forward model? Is the first meant to relate to data only?

Lines 117-118: Isn't the solubility model based on experimental data, and would therefore fit it well?

Line 123: Do you really mean "large" for the size of the eruption cloud? I can see that this would be related to the total SO₂ emissions, but would also be influenced by wind etc.

Lines 146, 164, 166: Shouldn't these reference Figure 2, not Figure 1?

Line 147: A couple of the r values may be quite a bit higher than 10. How to explain these?

Lines 186-190. A citation or two here might be appropriate.

Line 190: "Ins" -> "in"

Line 207: I found the text "magma oxidation state of 1 log unit above the nickel-nickel-oxide (NNO) buffer" a bit technical relative to the general audience at which this material is pitched (particularly compared to the introduction, which is nicely general).

Lines 234-225: This work is really not the "first to attempt to reconcile diverse geophysical and geochemical observations of volcanic eruptions from space"

Line 277: compressibility *of* the magma

Figure 2A: This figure shows erupted volume, sulfur yield, and r . But the 'raw' data must be erupted volume and reservoir volume change; from these, r is computed. So, it would be good to also show the estimated reservoir volume change. Also, the lowest y-axis value is 100. Finally, it looks like there are two values plotted from Puyehue (?).

Supplementary Figure 2: This figure shows porosity at what depth?

Response to reviews

We have increased our citations to 61. Word count is 3682, without methods; 4345 with methods.

Global changes:

“fluid” everywhere changed to “vapor” or “magmatic vapor”

Added to the acknowledgements, thanked reviewers.

We have added color symbols to Fig 4 to match Fig. 2 and added labels for source depth. We have changed the axes to log scale.

We have added additional discussion of Fig. 4 into the text.

We have added additional detail to Supplementary Table 1, adding some footnotes to explain which parts of the eruption our data refers to, and also included a new column to show source depth and geometry, to illustrate that the vast majority of the cases we have used are modelled using a MOGI source.

Reviewer #1

The paper describes a new approach to understand the occurrence and variation of gas-rich eruptions by linking two kinds (passive and active) of satellite observations with magmatic processes. The paper is clearly written and contains new material, presented in an informative and rigorous manner. The observation of very rich or poor yields of SO₂ gas from explosive volcanic eruptions has not previously been explained or investigated in an entirely satisfactory way. This paper provides some new insights into gas compressibility in the magma and a model that helps to understand the wide variation of SO₂ yields. The authors are careful to caveat their results and do not claim more than they can from the analysis of the data presented. This appears to be the first paper suggesting an approach that utilises satellite data in a quantitative manner to elucidate processes deep below the surface.

The model approach is interesting because it uses the observations to try to delineate magmatic processes without taking a formal mathematical approach of assimilating data or using mathematical inversion. I would suggest the more formal approach is premature as the data are not sufficiently accurate and the models are under development. This paper is a very timely and a novel contribution - it has all the hallmarks of originality and will be of great interests to those studying explosive volcanic activity. The authors themselves note that this work is a "first attempt" and I would anticipate that refinements will follow rapidly. The data and methodology are sound. The quality of presentation is also excellent and the Supplementary material I found very helpful.

There is not much discussion on uncertainties and I find this the only weakness in the paper. I think some statements on the uncertainties in model parameters and especially concerning the satellite data observational errors would be worthy of inclusion - perhaps in the Supplementary material(?) The conclusions are clear and robust and not extravagant. I have a feeling that the authors have a lot more interesting material to present and this is at an early stage of development. I assume that other papers are in preparation but that the need to publish their ideas quickly is paramount. The paper is therefore, in my view, very appropriate for Nature Communications.

References are appropriate and up-to-date (at least in the area of research that overlaps with mine). I would suggest one additional reference:

Kremser, S., et al. (2016), Stratospheric aerosol—Observations, processes, and impact on climate, *Rev. Geophys.*, 54, doi:10.1002/2015RG000511.

This is a new (review) paper and links nicely with this work because it shows why we need to know more about how and why volcanoes sometimes emit large amounts of SO₂ and why they are sometimes gas-poor.

1.1 We have added some discussion of uncertainties in the supplementary material, reproduced here:

“Uncertainties in satellite-based observations of SO₂ clouds

The observations of gas clouds collated for this study are from a number of different sensors on different platforms, but are dominated by observations by the Ozone Mapping Instrument (OMI) and the Atmospheric Infrared Sounder (AIRS)¹, which quantify vertical column densities of sulfur dioxide using its characteristic absorption signature in the UV and IR respectively¹⁻³. The observation of SO₂ from space using radiance data is subject to many sources of error and uncertainty⁴⁻⁶, which are summarised here. They may be grouped into uncertainties arising from sensor response (including the OMI “row anomaly”⁷), retrievals (e.g. interference with other gases, principally ozone³), local factors (e.g. cloud cover, interference by ash⁸), detection limits⁴, inherent features of plumes such as SO₂ loss by oxidation and dilution^{5,9}. In this study we have attempted to minimize errors by focussing only on larger, discrete explosive eruptions, where gas plumes are injected high into the troposphere and even into the stratosphere. Absolute errors are difficult to constrain, as other ground-based techniques to measure SO₂ are likewise subject to errors including those caused by excess scattering of UV radiation¹⁰ but attempts have been made to conduct calibration and comparison experiments using opportunistic observations of gas clouds from the ground and from space¹¹.

Uncertainties in geodetic estimates of volume change.

The observations of volume change in this study are collated from both satellite- and ground-based measurements of surface deformation, both of which are subject to inherent limitations and uncertainties. The ability of InSAR to measure surface deformation is limited by the availability of satellite images and the correlation between the images (the coherence of the interferogram). Coherence is limited by steep slopes, particularly for image pairs with high perpendicular baselines, and surface changes, which are particularly common in vegetated regions, for image pairs with long temporal separation and for interferograms created using short-wavelength instruments (X-band, C-band)¹². The uncertainty of the deformation measurement is dominated by the influence of tropospheric water vapour, and is particularly an issue for high relief volcanoes where atmospheric stratification causes systematic error correlated to topography¹³. Under favourable conditions, long, dense timeseries of InSAR data can give uncertainties better than 1mm/yr¹⁴, but for a single interferogram at tropical volcanoes, errors can be as much as 17 cm¹⁵. For ground-based networks such as GPS, the limitation is typically the number and location of instruments, particularly during large explosive eruptions as near-field sensors are often inoperable or destroyed.

The greatest source of uncertainty is the assumptions required to convert surface deformation into source volume changes. For simplicity, this paper is based on spherical sources within uniform, elastic half-spaces¹⁶, but in practice, many more parameters need to be considered, in particular topography and thermal and mechanical heterogeneities^{17,18}. Of particular relevance here, is the shape of the reservoir, which controls chamber compressibility. The compressibility of a spherical cavity is always less than that for an ellipsoidal cavity. For deep prolate sources, $\beta = 1/\mu$, so errors do not exceed 25%, but for extremely oblate cavities, approximating a circular crack, the difference can be two orders of magnitude^{19,20}. This contrast in compressibility has been applied to dyke intrusions to explain the discrepancy between the volume change of the source and sink²¹.”

We have added the following citation into the main paper at line 44:

Kremser S, Thomason LW, Hobe M, Hermann M, Deshler T, Timmreck C, Toohey M, Stenke A, Schwarz JP, Weigel R, Fueglistaler S. Stratospheric aerosol—Observations, processes, and impact on climate. *Reviews of Geophysics*. 2016 Jan 1.

Reviewer #2

This is a very interesting paper that uses physical and thermodynamic models to link measurements of surface subsidence accompanying eruptions with remotely sensed estimates of Sulfur emitted into the atmosphere. This is a great example of how merging multiple data types through models yields more information about quantities of interest than treating the data independently. The paper should be published with generally modest revisions.

There are a few main points:

1) It appears that the calculations assume a spherical chamber; it could be important to consider the effects of non-spherical sources. In the methods it is implied that the chamber compressibility holds for ellipsoidal (nonspherical) chambers, which has been stated in the literature, but is incorrect. Amoruso and Crescentini (2009) give results for volume change of ellipsoidal chambers. For example, shallow sill like chambers can give much larger volume changes per unit pressure change. I think it would be important to note how much of the variability in the ratio r , could be due to difference in chamber compressibility, since it is ultimately the ratio of magma to chamber compressibility that matters.

2.1 A discussion of the influence of chamber shape has been included in the discussion of uncertainties on geodetic measurements of volume change in the supplementary material (see response to reviewer 1, above) and we have incorporated this discussion into the main text also (underlined), at line 316:

“For elliptical chambers, or sill shapes, the expected volume change in response to a particular change in pressure is expected to be larger than for a spherical source, with the increase ranging from 1.1 to 1.3 times larger for a prolate spheroid with aspect ratio 1.5 to 1 (horizontal radius to height) to 5 times larger for a sill of aspect ratio 5 to 1¹⁹. If the ground deformation data drawn upon here (see Table 1, Supplementary data for citations) were inverted for elliptical magma reservoir sources, then the values for r , above (**Fig. 2**), would be reduced”

2) It would also be useful to have some sense how uncertainties in the thermodynamic data could influence estimates of volatile concentrations and oxidation state.

2.2 We have added the following to supplementary material:

“Uncertainties in thermodynamic modelling

The equilibrium constants for the redox reactions in the gas phase in the model Dcompress²² are calculated after²³. The solubility laws are constrained experimentally and are given in table 1 of²². The uncertainties within the model arise from exploring the less well constrained parts of the melt-C-O-H-S fluid solubility regime. In the examples we investigate here, we explore standard basaltic and rhyolitic melt compositions, for which there are numerous experimental studies of volatile solubility to draw on e.g.²⁴ and thus the model is well constrained for these compositions. Estimates of the uncertainties on melt volatile compositions are in general <10% for pressures >1 kbar²².”

3) I do not understand Equation (1) for magma compressibility. If ρ is the magma density then the compressibility is completely given by the derivative of density with regard to pressure, there is no other term for bulk modulus of melt. I thought that perhaps the authors were trying to use a mixture model to combine liquid and gas phase compressibilities, but this doesn't seem to be the case, and anyway would depend on the volume fraction of gas phase (porosity).

2.3 We made an error in the text, underlined below, now corrected. This does not affect the calculations – the error is in the text only. Beta is the bulk magma compressibility (melt + gas). In response to reviewer 3 we have also added an additional section to the methods, outlining in more detail each step of the calculations (see later in response to reviewer 3).

Line 465:

“The magma compressibility, β_{magma} , including that arising from this sulfur-bearing gas phase, is given by²⁵:

$$\beta_{magma} = \frac{1}{\rho} \frac{\partial \rho}{\partial p} \quad (1)$$

where β_{magma} is the bulk magma compressibility (Pa), ρ is magma density (from the outputs of the thermodynamic model, described above) and p is pressure. The volume removed during eruption is accommodated by deformation of the host rock and compressibility of the magma.”

4) Considering S outgassing (starting on line 242). Magma outgassing to the surface does not modify the results as long as it is magma that eventually reaches the surface. It doesn't matter when the vapor phase reaches the atmosphere. What would make a difference is if there is outgassing of magma that never reaches the surface. This would lead to higher sulfur yields for a given erupted volume of magma.

2.4 We agree with the reviewer and we have add the caveat, new line 264:

“Magmas might outgas sulfur but not erupt; this would lead to higher sulfur yields per unit volume of magma erupted.”

Minor Points:

Line 18 – bubbles cause magma to become “more” compressible. Even bubble free magmas are at least somewhat compressible.

2.5 Added “more” before compressible, line 18.

Line 33 – explain that “fringes” are lines of constant phase change

2.6 Modified sentence to “..typically displayed as “fringes”, which are contours of constant phase change.”

Beginning on Line 64 - what is "surface volume change"? I don't think you mean the volume of the surface subsidence. Please clarify. Also, does an eruption ever empty the chamber completely? I suspect not. I found this discussion somewhat confusing. What really matters is the ratio of magma to chamber compressibility.

Line 112: Define r earlier and more thoroughly.

2.7 We have deleted the part about the emptying of the magma reservoir and modified this substantially, adding the underlined words below (see also tracked changes). We have inserted a definition of r above the preceding paragraph:

New line 82:

“We may define a ratio, r , between the erupted volume (V_e) and volume change of the subsurface reservoir (ΔV_c), given by²¹ $r = \frac{V_e}{\Delta V_c}$ Three theoretical possibilities can be

envisioned for a spherical source: 1) If the magma is considered incompressible, the reservoir volume change would be equal to the erupted volume ($r = 1$) and the surface volume change will actually be greater due to compression of the host rock; ii) for a gas-free but slightly

compressible magma, the volume of the surface subsidence might be approximately equal to the erupted volume but the reservoir volume change will always be less ($r > 1$); and iii) an exsolved volatile phase in the magma will increase its compressibility by an order of magnitude, thus the eruption of a significant volume of material is accommodated by expansion of the remaining magma causing little volume change of the chamber at depth or deformation at the surface²⁶⁻²⁹ ($r \gg 1$). A poor understanding of the total mass fraction of exsolved vapor however, leads to a lack of constraint on inferred volume changes from InSAR modelling. It is worth noting that assumptions about geometry and material property are required in order to infer subsurface volume change from the surface displacements observed by geodetic techniques such as GPS and InSAR. As our goal is to establish a framework for reconciling observations of gas and deformation, we consider the simplest geometry - that of a spherical source - and use derived values of volume change from published studies. It is our intention that future studies refine this framework for specific circumstances, and the influence of these assumptions are discussed in the supplementary material..”

Figure 3: why is color scale for partition coefficient in part B different from A and C?

2.8 The fluid-melt partition coefficients for sulfur increase to much higher values for these examples, and so we used the same range in color to encompass a much larger range in partition coefficient.

Line 118: “the mass of sulfur expected to be present in the co-eruptive atmospheric cloud per 1 km³ magma ranges over several orders of magnitude”. But this is only true for variations in Oxygen fugacity. Water and CO₂ content don't give orders of magnitude change, although you do say that later.

2.9 We agree and have modified the sentence, new words underlined.

New line 188:

“Partitioning of sulfur into fluid in the pre-eruptive magma reservoir varies over orders of magnitude for the conditions studied (particularly upon varying the water content of the melts and the oxygen fugacity) and in all cases decreases with depth, with partition coefficients comparing well to those from experimental studies^{30,31}.”

line 159 porosity permeability relationship is non-linear even if no crystals are present (e.g., Carmen-Kozeny relation)

2.10 Agreed, changed wording.

New line 254:

“Experiments using analog materials have shown that crystals significantly alter the bulk rheological properties of magmas, leading to a range of gas transport behaviours involving fingering and quasi-brittle fracturing^{32,33}.”

Lines 164 and 166. Do you mean to be referring to Figures 4 and 2B? I was confused.

Line 284. Eq (2) is messed up in my version, although I assume its correct.

2.11 These should be Fig 2 and Fig 2B. Now corrected.

Reviewer #3

In this work, the authors compute the compressibility ratios 'r' for a number of eruptions using estimates of erupted volume and reservoir volume change from the literature. Using the solubility package D-Compress, and making assumptions about reservoir compressibility, magma compressibility is related to volatile concentrations. Assuming degassing during eruption, these estimates are related to observations of sulfur emissions.

This work is a valuable contribution to the literature and the authors are to be commended for working to integrate disparate observations of volcanic eruptions. Quantitatively linking observed gas emissions with subsurface processes is an important problem and should inspire future work. There are however some important issues that need to be addressed prior to publication.

These issues are described below.

General Comments

=====

(1) I do not believe that Equation (1) is correct. The compressibility of a multiphase melt may be computed as the sum of the individual compressibilities multiplied by their respective volume fractions (see for instance Mastin et al. 2008, equation 6). Equation (1) here shows the bulk magma compressibility as the sum of melt (gas-free) and gas phases, but does not account for their relative abundances. The cited reference for this expression is Huppert and Woods 2002, but that paper (Equation 5) actually shows a sum of rock and magma compressibility, which is not the same as summing individual phase compressibilities in the magma. Since gas volume fractions are much lower than melt volume fractions, it seems this error could have a significant influence on results presented in the manuscript.

See response to reviewer 2, point 2.3 above.

We have accounted for the relative abundances of the gas and melt phases in the calculation of bulk magma compressibility, beta above. We calculate, for each pressure step, the mass (and moles) and composition of the gas phase in equilibrium with the melt, and then calculate bulk magma density by summing the volumes of the gas components (using the ideal gas law) and the volume of the melt fraction and the total mass. So the components of the gas and melt phases are implicitly taken into account.

Have added, line 458:

“Also arising from the gas phase calculations are a mass of vapor per unit volume of magma. Using the molar proportions of H₂O, CO₂ and S gases in the vapor combined with their molar masses, it is possible to calculate the total mass of vapor associated with 1 m³ magma at pressure increments. The molar volumes of the vapor may then be used to calculate a bulk magma density, ρ , at each pressure step, using the Ideal Gas Law and the bulk density may be calculated using an appropriate density for the melt proportion (2400 kgm⁻³ for rhyolite; 2800 kgm⁻³ for basalt).”

(2) There are some problems with the way that reservoir compressibility beta_c is handled in this work. First, there is a minor confusion on Line 283, which describes beta_c as host rock compressibility, when it is in fact reservoir compressibility (these are not the same). More importantly, chamber compressibility is given as 3/4mu. However, this expression is only strictly correct for a spherical reservoir in a full-space. In other cases, compressibility deviates from 3/4mu. Analytical expressions for certain end member cases can be found in Amoroso and Crescentini (2009), and numerical computations for chambers of different shapes and depths can be found in Anderson and Segall (2011). For shallow, sill-like chambers, results can be very different than 3/4mu. Interestingly, an expression for an opening dyke is given on Line 284, but it is not at all clear how this expression is used in the work. Geodetic references used in this study should be consulted to ensure that the assumption of a near-spherical, relatively deep reservoir is actually valid -- or, if the expression for an opening dike is used, that must be clarified.

We have now corrected the definition of beta_c, see response to reviewer 2, points 2.3, 2.1.

Finally, some accounting must be made of the great deal of uncertainty associated with host rock rigidity, to which reservoir compressibility is directly related. Altogether, the authors seem to assume that reservoir compressibility is known, when in fact it may be only very poorly known. At the least, some discussion about this assumption is warranted. Clearly, for a very bubbly magma the reservoir compressibility will not be important, but the minima of 1.4 mentioned in the text could certainly be affected.

We agree. Have added the following section into the methods, line 490:

“The shear modulus of crust, μ , in volcanic areas is not well known and ranges from ~ 0.1 GPa for very compliant rocks to 30 GPa for non-compliant, or stiff rocks^{21,34}. Compressibility of

degassed basalts, β_{magma} , at crustal depths is in the range 0.6–1.0 10^{-10} Pa⁻¹, resulting, in theory, in a value of r for degassed magma ranging from 1.05 to 9, which encompasses much of the range in the natural data. Improved constraints on the compliance of the crust are clearly needed to refine these models in the future. An equally or perhaps more important control on the range in r is the variability in the value of β_{magma} , which we predict will extend up to 30^{-10} Pa⁻¹ for volatile-rich magmas stored at depths of > 3 km, yielding a large range in r , up to 15 for realistic crustal parameters and magma depths.”

(3) There are some problems with the way deformation modeling is handled, or at least discussed. First, the relationship between observed deformation, reservoir volume change, and estimates of 'r' are a bit unclear. As far as I can tell, surface volume change and reservoir volume change are being confused, at the very least in the text (it's not clear if computations are influenced). For instance, Line 58 refers to "the magnitude of the subsidence signal", while just a little later on Line 69 it is "volume change of the chamber at depth" that is discussed. The ratio r is computed from equation 2 using reservoir volume change, not surface volume change. Although volume of surface uplift or subsidence (integrated vertical displacements) is also a metric used in deformation studies (such as the Johnson 2000 paper referenced here), it is not generally the same as reservoir volume change. Also, later, on Line 278, ΔV_c is defined as "change in volume observed by InSAR". It is important to recognize that ΔV_c is change in *reservoir* volume, and it is not observed by InSAR, but is rather modeled from InSAR data. These points need to be clarified in the text, and the meaning of ΔV_c more clearly stated.

We agree with the reviewer that there is a confusion in the text. The literature data in table 1 is for volume changes of the modelled source, NOT integrated surface volume change. We have gone through the manuscript to make sure that this is consistent throughout. Specifically, we have changed "change in volume observed by InSAR" as picked out by the reviewer above, to "the change in volume of a spherical magma reservoir (the "Mogi" source¹⁶, constrained by observed by InSAR or GPS (ΔV_c))."

I also found the discussion on lines 62-70 a bit problematic. These statements about compressibility are well known, and assertions in i–iii need to be backed up with either specific citations or equations. Also, I suspect that reference 14 (Mastin) is not the one intended.

We have now modified this paragraph substantially – see response to reviewer 2 above, point 2.7. We have replaced the Mastin citation.

A final point is that the manuscript focuses on the use of space-based observations, but some geodetic inversions utilized by the cited references likely include other observations as well (for instance, the Sigmundsson paper also uses GPS data to derive subsurface volume change).

Noted and we have added a statement to say that some of the observations are groundbased. This has no impact on the science in this paper. Fig 2 figure caption:
"The deformation data are derived from InSAR studies as well as ground-based GPS networks."

(4) The novelty of magma/reservoir compressibility and constraining a reservoir's exsolved volatile content is, I feel, a bit over-stated. The abstract states that "most eruptions are associated with ground deformation much less than expected for the erupted volume", but this is not really true -- the difference between erupted volumes and inferred reservoir volume change, and relation to magma compressibility, has been well known for quite some time (in other words, no one should really expect these things to generally be equal).

The data show that the ratio r is highly variable. Some eruptions show little discrepancy; some show extremely muted ground deformation. We would argue that this variability remains poorly understood. We have not changed this sentence.

The difference between this study and previous work is that we try to link two sets of observations - ground deformation and gas emissions. In previous work, the volatiles were

implicated, but there was no independent way to constrain the amount of water and CO₂ in the magma reservoir. We are introducing a new approach –using the sulfur content of the contemporaneous gas cloud to back out information about the volatile phase, allowing the two datasets to be interpreted in tandem and in doing so tell us something about the importance of the exsolved fluid phase and the conditions under which magmas were stored prior to eruption.

This work is the first to present a framework within which both SO₂ emissions and ground deformation may be considered together for a range of different eruptive scenarios and taking a global view. It builds on our recent advances in understanding the thermodynamic controls on sulfur partitioning into fluids, and on recent shifts in our view of magma reservoir form and architecture, including the role of crystal mushes might play in gas storage and transport. Previous papers used very simple empirical solubility laws to investigate CO₂ and H₂O outgassing from magmas, and none considered how sulfur may behave in such a system. We present a study here aimed at picking out features of the global dataset, which is entirely novel, as the other two reviewers note.

Some discussion of previous work here is warranted (references include Johnson 1987, 1992, and 2000, Rivalta and Segall 2008, etc.); also, Line 60 should include some references to previous examples of the phenomenon being discussed. Previous workers have also directly modeled reservoir volatile content using the ratio r , as done here. For instance, both Mastin et al 2008 and Anderson and Segall 2013 use erupted volume and ground deformation data at Mount St. Helens, together with H₂O + CO₂ solubility models, to estimate exsolved water and CO₂ in the reservoir. This is precisely the approach followed in this manuscript, with the additional incorporation of SO₂ emissions data and a model that includes sulfur solubility.

We have already cited Rivalta and Segall, 2008 and Johnson 2000. We have added the additional Johnson citations suggested above into this section (see tracked changes).

See point above for our response to previous work and how this study advances understanding.

(5) As noted in the manuscript, reservoir compressibility is dominated by exsolved water, not SO₂. Thus, estimates of r yield mostly constraint on water content. Constraint on sulfur, CO₂, and NNO must come from solubility modeling and observed SO₂ emissions. This would seem to be a difficult problem, particularly when considering the common occurrence of "excess" sulfur in the reservoir. In other words, given all these adjustable parameters, isn't it perhaps almost too easy to reproduce observations? I understand that in the case of very low sulfur emissions (Chaiten), the lack of excess sulfur simplifies the problem, but what's less clear to me is what can be learned in the more common case (Okmok). For instance, it's noted that the given parameters are reasonable for an arc magma -- but would "unreasonable" parameters also fit the data? Some further discussion would be very helpful here.

We make it very clear that the dominant magmatic volatiles that cause magma compressibility are water and/or CO₂ (around line 88). As a community, we now understand very well however, how sulfur partitions into this volatile phase (through experiments and through modelling), making it now possible to relate magma compressibility and bulk volatile content to sulfur outgassing (the "difficult problem" the reviewer mentions above), which is exactly the novel thrust of this paper. Understanding the controls on sulfur partitioning and on deformation in tandem is the fundamental advance here.

All models have adjustable parameters. The beauty of this model framework is that it now becomes easily testable. If we were to conclude that a certain set of observations (deformation, gas) at the surface could be explained either by, say, a very water rich magma; or by an extremely oxidised magma, we could then take those hypotheses and test them using focussed petrological and geochemical studies and also bringing in other geophysical datasets, including depths derived from deformation observations. This work is intended to establish a quantitative framework for understanding diverse sets of data and to allow the many different kinds of observations and models to be fitted together.

The reviewer makes an important point about reasonable and unreasonable parameters. The answer in general is that unreasonable parameters (e.g. >10 wt% melt water contents; an fO_2 of >3) would not fit our observations at all. We do not explicitly show this, but for extremely high water contents magmas would be very highly compressible, generating r values much higher than observed; and for extreme magma fO_2 sulfur would barely degas, with much of it becoming locked away in anhydrite at depth.

Also, Figure 2B shows erupted sulfur vs. sulfur yield, and the text notes some correlation. But isn't sulfur yield computed directly from erupted sulfur and the erupted volume? In that case, is the correlation observed not simply related to the erupted volume ($T_g/km^3 / T_g = 1/km^3$)? Am I missing something?

The reviewer would be correct in thinking this was a facile result if the erupted volume (denominator) is a constant. If the amount of sulfur outgassed to the atmosphere was directly proportional to the amount of magma erupted, one would expect the sulfur yield (kg S per km^3 magma erupted) to be constant across all eruptions. This is clearly not what is observed. What is observed is that the very largest sulfur clouds are generated in those eruptions where there is the greatest amount of "excess sulfur"; and that the sulfur yield is not proportional to the erupted volume. This complexity is part of the problem that we attempt to address in this paper. Understanding which eruptions produce the largest sulfur yields to the atmosphere is not just a case of picking out the largest eruptions. There are a critical set of conditions that are more likely to yield sulfur-producing eruptions (and these are the eruptions that are preferentially preserved in ice core records of course).

(6) Some clarification of some of the modeling work would be very helpful. First, are sulfur emissions set as equal to the exsolved phase only? Lines 265-269 seem to imply that these emissions are included, but Lines 241-243 seems to imply that they are not.

The sulfur emissions from the model are the sum of the sulfur present in the exsolved fluid prior to eruption. The emissions do not include sulfur that has outgassed during magma ascent to the surface.

Also, for Figure 3A, can you explain why sulfur yield decreases with increasing water content? Since D-Compress is a bit of a black box, this relationship is not immediately obvious (to me, anyway).

The chemical model Dcompress²² is based on the core premise of equilibrium such that that for volatile species dissolved in a fluid-saturated melt, equilibrium conditions impose that the fugacity, f_i , of species i in the gas phase equals that in the melt³⁵. Increasing the melt water contents whilst keeping the sulfur concentrations the same in the melt results in a decrease in the fugacity of sulfur in the melt decreases, requiring a corresponding decrease in the f_{S_2} (or you can think of it as the mass proportion of sulfur) of the magmatic vapor phase. This is why increasing the water content of the melt decreases the sulfur yield. It is probably non-linear however – as you increase water content further, the larger volume of the fluid takes over, despite the lower concentration of sulfur in the vapor phase, so that there is a minimum amount of sulfur associated with a particular range in water contents.

We have discovered a slight error here in the figure. The highest water content model run we used a slightly higher fO_2 , as we considered this to be closer to reality, based on the observation that oxidation state of magmas correlates closely with melt water content³⁶ (there is a common reason for both – transfer of slab fluids). Have inserted the following into the caption to explain and have modified the figure slightly by adding an asterisk.

"*In panel A, the oxygen fugacity is NNO+1 for the model runs with a melt water content of 6 and 8 wt% H_2O and NNO+2 for the model run with a melt water content of 10 wt%, justified by global observations³⁶."

I also find Figure S3 quite confusing. Is the idea that only certain combinations of NNO, H_2O , and CO_2 fit the data? That's great, but I'm confused by the dashed lines vs. solid dots.

The solid dots represent the modelling results, and the dashed lines delineate the regions of the plot (and S yield and ranges in r) that would result from melts with the labelled water contents e.g. in the left hand plot melts with 5 wt% water would produce ranges in S yield and r value at the top of the plot, above the top dashed line; melts with 3 wt% water would produce intermediate r values and a range in S yields shown by the region between the top and middle dashed lines; and water-poor melts would produce the range in r and S yield given by the lowermost region of the plot, below the bottom dashed line. The actual observations are shown by the heavy black line, which allows estimation of the magma's volatile contents, which might then be tested using melt inclusion studies, for example. In the case of Okmok, the observations are consistent with a water content of ~ 3 wt%, which is borne out by petrological studies by Larsen et al. We have modified the figure caption to make this clearer (see tracked changes).

Other comments

=====

Line 21: A bit unclear. How can you predict surface deformation and other observations without knowing the volume of erupted products, etc.? Certainly, reservoir depth, oxidation state, and volatile content are inadequate to predict surface deformation.

Agreed - poor wording. Changed to:

"Here we present a model that allows the relative magnitude of the surface deformation (compared to the volume of magma erupted) and the sulfur mass loading into the atmosphere to be predicted as a function of magma reservoir depth, magma oxidation state and the total volatile content of the magma, which explains the array of natural data"

Line 33: "typically displayed as 'fringes'" is potentially confusing in the main text. Consider removing or moving to the appropriate caption.

Have modified in response to reviewer 2:

"..typically displayed as "fringes", which are contours of constant phase change."

Line 37: I think there is a big difference between a pre- and syn-eruptive lack of deformation. In the latter case, material is clearly leaving the reservoir. But it is not necessarily clear that magma is entering the reservoir prior to eruption.

We agree that the reasons for a lack of both types of deformation might be varied. We are drawing attention to the fact that enhanced magma compressibility may cause both to be muted.

Lines 62-70: I find these three possibilities a bit confusingly described. Can this be simplified?

We have modified this paragraph in response to reviewer 2, see above.

Line 71: Modeling of deformation data yields volume changes, but pressure change cannot be inferred independently of volume. Is the inference here that pressure change can be inferred from the solubility modeling? If so, please clarify and/or expand this.

No this is not the implication - note our use of the word "or". To be clear, have deleted "or pressure".

Line 95: "there have been very few attempts". Please provide citations. Line 97: "even fewer reconciling". Please provide citations.

One study has linked ground deformation with sulfur emissions (as a proxy for a gas phase), but not quantitatively³⁷ and several studies consider CO₂ and H₂O (cited above). So we have modified the sentence thus (changes underlined):

Line 108: 'Now a large database of sulfur emissions from volcanoes exists⁶, but there have been very few attempts³⁷ to fit these observations into this now well-established theoretical framework for sulfur gas-melt partitioning and none reconciling the geochemical features of the magmatic

vapor with the consequences for magma compressibility, yet this approach clearly has enormous potential for strengthening our constraints on pre-eruptive magma storage conditions.”

Lines 101-106: I don't really understand the difference between these two items. How can you explore the sensitivity if you don't have a forward model? Is the first meant to relate to data only?

Agreed; this is not clear. We have modified this paragraph to:

“Here we present a novel approach aimed at creating a model to predict space-based observations of atmospheric sulfur mass loading of sulfur and the relative magnitude of observed ground deformation during discrete explosive volcanic eruptions. In doing so, the sensitivity of ground deformation and gas emissions to the intrinsic magma storage and bulk volatile contents of the magma will be explored.”

Lines 117-118: Isn't the solubility model based on experimental data, and would therefore fit it well?

The comparison of model fluid-melt partition coefficients to experiments is a check that the model is producing realistic results. The experimental data in the literature is slightly more diverse than that used as calibration of the model.

Line 123: Do you really mean "large" for the size of the eruption cloud? I can see that this would be related to the total SO₂ emissions, but would also be influenced by wind etc.

We mean large in mass, not volume. Modified (words added underlined):

“The H₂O and CO₂ contents of the magma, as well as the oxidation state, are all important for controlling the partitioning of sulfur into a fluid phase in the crustal magma reservoir prior to eruption and hence determining how large (in mass) the co-eruptive sulfur cloud that is observed from space is.”

Lines 146, 164, 166: Shouldn't these reference Figure 2, not Figure 1?

Corrected

Line 147: A couple of the r values may be quite a bit higher than 10. How to explain these?

We attempt to show the vectors influencing the r value and sulfur yield in figure 4. Shallower chambers and higher water contents would allow high values of r , as would segregated regions of gas-rich magma at the roof of the magma reservoir.

Lines 186-190. A citation or two here might be appropriate.

New line 230, inserted citations:

“The deformation of the country rocks in response to volcanic eruptions (assuming that the response of the country rocks themselves do not vary) is expected to be most muted (i.e. high r) for shallow magma reservoirs, for very water-rich magmas, or for gas-rich magma reservoir caps^{20,21}.”

Line 190: "Ins" -> "in"

corrected

Line 207: I found the text "magma oxidation state of 1 log unit above the nickel-nickel-oxide (NNO) buffer" a bit technical relative to the general audience at which this material is pitched (particularly compared to the introduction, which is nicely general).

Changed to “NNO+1”

Lines 234-225: This work is really not the "first to attempt to reconcile diverse geophysical and geochemical observations of volcanic eruptions from space"

Previous studies have considered the role of volatiles in magma compressibility, but this paper is the first to use SO₂ observations from satellites with ground deformation measurements in a combined model. We are not aware of another paper that has done this. We have cited all of the papers this reviewer suggests (see above).

Line 277: compressibility *of* the magma

Corrected.

Figure 2A: This figure shows erupted volume, sulfur yield, and r . But the 'raw' data must be erupted volume and reservoir volume change; from these, r is computed. So, it would be good to also show the estimated reservoir volume change. Also, the lowest y-axis value is 100. Finally, it looks like there are two values plotted from Puyehue (?).

The reservoir volume change is shown in supplementary table 1.

Fig 2 corrected ("100" and deleted lower data point, which was an error; see table 1 supp); figure 4 "Cordon Caulle" changed to "Puyehue" to make consistent.

Supplementary Figure 2: This figure shows porosity at what depth?

The depth is a function of r (which is the observation), so this plot shows a range of depths 2-10 km. The shallower depths will be at higher r . This plot is to indicate what sort of magma gas fractions might be expected in the magma reservoir, which which may range over a range of depths.

- 1 Prata, A. & Bernardo, C. Retrieval of volcanic SO₂ column abundance from Atmospheric Infrared Sounder data. *Journal of Geophysical Research: Atmospheres* **112** (2007).
- 2 Yang, K. *et al.* Retrieval of large volcanic SO₂ columns from the Aura Ozone Monitoring Instrument: Comparison and limitations. *Journal of Geophysical Research: Atmospheres* (1984–2012) **112** (2007).
- 3 Krotkov, N., Carn, S., Krueger, A. J., Bhartia, P. K. & Yang, K. Band residual difference algorithm for retrieval of SO₂ from the Aura Ozone Monitoring Instrument (OMI). *Geoscience and Remote Sensing, IEEE Transactions on* **44**, 1259-1266 (2006).
- 4 Carn, S., Krotkov, N., Yang, K. & Krueger, A. Measuring global volcanic degassing with the Ozone Monitoring Instrument (OMI). *Geological Society, London, Special Publications* **380**, 229-257 (2013).
- 5 McCormick, B. T. *et al.* A comparison of satellite and ground-based measurements of SO₂ emissions from Tungurahua volcano, Ecuador. *Journal of Geophysical Research: Atmospheres* **119**, 4264-4285 (2014).
- 6 Carn, S., Yang, K., Prata, A. & Krotkov, N. Extending the long-term record of volcanic SO₂ emissions with the Ozone Mapping and Profiler Suite nadir mapper. *Geophysical Research Letters* **42**, 925-932 (2015).
- 7 McCormick, B. T. *et al.* Volcano monitoring applications of the Ozone Monitoring Instrument. *Geological Society, London, Special Publications* **380**, 259-291 (2013).
- 8 Corradini, S., Merucci, L. & Prata, A. Retrieval of SO₂ from thermal infrared satellite measurements: correction procedures for the effects of volcanic ash. *Atmospheric Measurement Techniques* **2**, 177-191 (2009).
- 9 Lee, C. *et al.* SO₂ emissions and lifetimes: Estimates from inverse modeling using in situ and global, space-based (SCIAMACHY and OMI) observations. *Journal of Geophysical Research: Atmospheres* **116** (2011).
- 10 Kern, C. *et al.* Radiative transfer corrections for accurate spectroscopic measurements of volcanic gas emissions. *Bull Volcanol* **72**, 233-247 (2010).
- 11 Carn, S. & Lopez, T. Opportunistic validation of sulfur dioxide in the Sarychev Peak volcanic eruption cloud. *Atmospheric Measurement Techniques* **4**, 1705-1712 (2011).
- 12 Ebmeier, S., Biggs, J., Mather, T. & Amelung, F. Applicability of InSAR to tropical volcanoes: insights from Central America. *Geological Society, London, Special Publications* **380**, 15-37 (2013).
- 13 Parker, A. L. *et al.* Systematic assessment of atmospheric uncertainties for InSAR data at volcanic arcs using large-scale atmospheric models: Application to the Cascade volcanoes, United States. *Remote Sensing of Environment* **170**, 102-114 (2015).
- 14 Hammond, W. C., Blewitt, G., Li, Z., Plag, H.-P. & Kreemer, C. Contemporary uplift of the Sierra Nevada, western United States, from GPS and InSAR measurements. *Geology* **40**, 667-670 (2012).

- 15 Heleno, S. I. *et al.* Seasonal tropospheric influence on SAR interferograms near the ITCZ—The case of Fogo Volcano and Mount Cameroon. *Journal of African Earth Sciences* **58**, 833-856 (2010).
- 16 Mogi, K. Relations between the eruptions of various volcanoes and the deformations of the ground surfaces around them. (1958).
- 17 Masterlark, T. Finite element model predictions of static deformation from dislocation sources in a subduction zone: sensitivities to homogeneous, isotropic, Poisson-solid, and half-space assumptions. *Journal of Geophysical Research: Solid Earth* **108** (2003).
- 18 Hickey, J., Gottsmann, J., Nakamichi, H. & Iguchi, M. in *EGU General Assembly Conference Abstracts*. 13444.
- 19 Amoruso, A. & Crescentini, L. Shape and volume change of pressurized ellipsoidal cavities from deformation and seismic data. *Journal of Geophysical Research: Solid Earth* **114** (2009).
- 20 Anderson, K. & Segall, P. Physics-based models of ground deformation and extrusion rate at effusively erupting volcanoes. *Journal of Geophysical Research: Solid Earth* **116** (2011).
- 21 Rivalta, E. & Segall, P. Magma compressibility and the missing source for some dike intrusions. *Geophysical Research Letters* **35**, L04306, doi:10.1029/2007GL032521 (2008).
- 22 Burgisser, A., Alletti, M. & Scaillet, B. Simulating the behavior of volatiles belonging to the C–O–H–S system in silicate melts under magmatic conditions with the software D-Compress. *Computers & Geosciences* **79**, 1-14 (2015).
- 23 Ohmoto, H. & Kerrick, D. Devolatilization equilibria in graphitic systems. *American Journal of Science* **277**, 1013-1044 (1977).
- 24 Lesne, P. *et al.* Experimental simulation of closed-system degassing in the system basalt–H₂O–CO₂–S–Cl. *Journal of Petrology*, egr027 (2011).
- 25 Huppert, H. E. & Woods, A. W. The role of volatiles in magma chamber dynamics. *Nature* **420**, 493-495 (2002).
- 26 Mastin, L. G., Roeloffs, E., Beeler, N. M. & Quick, J. E. Constraints on the size, overpressure, and volatile content of the Mount St. Helens magma system from geodetic and dome-growth measurements during the 2004-2006+ eruption. *US Geological Survey professional paper*, 461-488 (2008).
- 27 Johnson, D. J., Sigmundsson, F. & Delaney, P. T. Comment on "Volume of magma accumulation or withdrawal estimated from surface uplift or subsidence, with application to the 1960 collapse of Kīlauea volcano" by PT Delaney and DF McTigue. *Bull. Volcanol* **61**, 491-493 (2000).
- 28 Johnson, D. J. Elastic and inelastic magma storage at Kīlauea volcano. *Volcanism in Hawaii* **2**, 1297-1306 (1987).
- 29 Johnson, D. J. Dynamics of magma storage in the summit reservoir of Kīlauea volcano, Hawaii. *Journal of Geophysical Research: Solid Earth (1978–2012)* **97**, 1807-1820 (1992).
- 30 Zajacz, Z., Candela, P. A., Piccoli, P. M. & Sanchez-Valle, C. The partitioning of sulfur and chlorine between andesite melts and magmatic volatiles and the exchange coefficients of major cations. *Geochimica et Cosmochimica Acta* **89**, 81-101 (2012).
- 31 Scaillet, B., Clément, B., Evans, B. W. & Pichavant, M. Redox control of sulfur degassing in silicic magmas. *Journal of Geophysical Research: Solid Earth (1978–2012)* **103**, 23937-23949 (1998).
- 32 Parmigiani, A., Huber, C. & Bachmann, O. Mush microphysics and the reactivation of crystal-rich magma reservoirs. *Journal of Geophysical Research: Solid Earth* **119**, 6308-6322 (2014).
- 33 Oppenheimer, J., Rust, A., Cashman, K. & Sandnes, B. Gas migration regimes and outgassing in particle-rich suspensions. *Front. Phys.* 3: 60. doi: 10.3389/fphy (2015).
- 34 Gudmundsson, A. The effects of layering and local stresses in composite volcanoes on dyke emplacement and volcanic hazards. *Comptes Rendus Geoscience* **337**, 1216-1222 (2005).
- 35 Scaillet, B. & Pichavant, M. A model of sulphur solubility for hydrous mafic melts: application to the determination of magmatic fluid compositions of Italian volcanoes. *Annals of Geophysics* (2005).
- 36 Kelley, K. A. & Cottrell, E. Water and the oxidation state of subduction zone magmas. *Science* **325**, 605-607 (2009).
- 37 Jay, J. *et al.* Locating magma reservoirs using InSAR and petrology before and during the 2011–2012 Cordón Caulle silicic eruption. *Earth and Planetary Science Letters* **395**, 254-266 (2014).

Reviewer #2 (Remarks to the Author):

I am satisfied that the authors have thoroughly responded to the extensive comments. Thus, I recommend publication at this time.

Reviewer #3 (Remarks to the Author):

Review of revised manuscript: "Observing eruptions of gas-rich, compressible magmas from space", by McCormick-Kilbride et al.

In the revised version of this manuscript the authors have worked to address reviewer concerns. General technique and conclusions did not require revision and remain unchanged. The manuscript will make a valuable addition to the literature and is ready for publication. I note only one area requiring further attention. Specifically, I still find that the paragraph beginning on Line 49 needs some clarification. The text continues to refer to "the magnitude of the subsidence signal" (Line 58). As pointed out by reviewers 2 and 3, the meaning of "magnitude of the subsidence signal" is not at all clear, nor is this probably what is really meant (volume change of the reservoir, presumably). Then, "surface volume change" is used on Line 67. What does this mean? Then, on Line 68, there is a new sentence: "the volume of the surface subsidence might be approximately equal to the erupted volume but the reservoir volume change will always be less ($r > 1$)". Here, both surface volume change and reservoir volume change are mentioned. Either the relationship between these things needs to be clarified or -- better yet -- "surface volume change" and "magnitude of the subsidence signal" removed entirely (as far as I can tell, they are not used elsewhere).

Response to final review

EDITORIAL REQUESTS:

* Nature Communications uses a transparent peer review system, where for manuscripts submitted from January 2016 we are publishing the reviewer comments to the authors and author rebuttal letters of our research articles online as a supplementary peer review file. Please let us know in the cover letter when submitting the final version of your manuscript whether you are opting out of this scheme or not.

This is fine.

* Please also review the changes in the attached copy of your manuscript, which has been edited for style, and address the comments and queries I have added. If using Word, please use the 'track changes' feature to make the process of accepting your manuscript more efficient.

We used the editor's commented version as the basis for the final version and accepted all of these changes.

We have also changed layout, reduced words in intro, moved a section to beginning of results, made extra subheadings in methods and results. We modified figures 1 and 2 (added panel labels, increased size of plots on RHS in fig 2).

We have swapped figures 2 and 3 in order, to avoid having to cite figure 2 in the introduction section. We suggest, however, that citing figure 1 (parts A and B) in the intro is appropriate because it contains purely introductory material.

We reduced the total number of citations to 70.

* Data availability statements and data citations policy: All Nature Communications manuscripts must include a section titled "Data Availability" at the end of the Methods section or main text (if no Methods).

We have inserted the following section:

"Data Availability

All of the data used in this paper have been published in the peer reviewed literature and are given in the supplementary material, table 1, which also contains the primary references for each dataset."

* Please check whether your manuscript contains third-party images, such as figures from the literature, stock photos, clip art or commercial satellite and map data. We strongly discourage the use or adaptation of previously published images, but if this is unavoidable, please request the necessary rights documentation to re-use such material from the relevant copyright holders and return this to us when you submit your revised manuscript.

Fig 1 shows two images at top that have been published in JGR (Solid Earth and Atmospheres respectively). I have obtained permission to use these in this article using their Rightslink online service (and paid the fees for both). The licences are uploaded as supporting files.

* Your paper will be accompanied by a two-sentence editor's summary, of between 250-300 characters (including spaces), when it is published on our homepage. Could you please approve the draft summary below or provide us with a suitably edited version.

Our suggestion below:

Satellite observations are an important tool in volcano monitoring, but observations such as ground deformation and gas emissions are treated independently. Here, the authors present a model coupling them through their link to magma volatile contents and storage depths prior to

eruption.

REVIEWERS' COMMENTS:

Reviewer #2 (Remarks to the Author):

I am satisfied that the authors have thoroughly responded to the extensive comments. Thus, I recommend publication at this time.

Reviewer #3 (Remarks to the Author):

Review of revised manuscript: "Observing eruptions of gas-rich, compressible magmas from space", by McCormick-Kilbride et al.

In the revised version of this manuscript the authors have worked to address reviewer concerns. General technique and conclusions did not require revision and remain unchanged. The manuscript will make a valuable addition to the literature and is ready for publication.

I note only one area requiring further attention. Specifically, I still find that the paragraph beginning on Line 49 needs some clarification. The text continues to refer to "the magnitude of the subsidence signal" (Line 58). As pointed out by reviewers 2 and 3, the meaning of "magnitude of the subsidence signal" is not at all clear, nor is this probably what is really meant (volume change of the reservoir, presumably). **Line 58, changed to:**

"The inferred volume change of the reservoir observed syn-eruption, however, is often many times less than the volume of magma erupted at the surface..."

Then, "surface volume change" is used on Line 67. What does this mean?

Line 67:

Changed to:

"reservoir volume change"

Then, on Line 68, there is a new sentence: "the volume of the surface subsidence might be approximately equal to the erupted volume but the reservoir volume change will always be less ($r > 1$)". Here, both surface volume change and reservoir volume change are mentioned. Either the relationship between these things needs to be clarified or -- better yet -- "surface volume change" and "magnitude of the subsidence signal" removed entirely (as far as I can tell, they are not used elsewhere).

We agree and have removed these terms entirely. Changed line 68 to:

"Three theoretical possibilities can be envisioned for a spherical source: firstly, if the magma is considered incompressible, the reservoir volume change would be equal to the erupted volume ($r = 1$). Secondly, for a gas-free but slightly compressible magma, the reservoir volume change will be less than the erupted volume ($r > 1$). Thirdly, an exsolved volatile phase in the magma will increase its compressibility by an order of magnitude, thus the eruption of a significant volume of material is accommodated by expansion of the remaining magma causing little volume change of the reservoir^{16-18,24} ($r \gg 1$)."